# 🍺 Inference-Time Policy Adapters (IPA): Tailoring Extreme-Scale LMs without Fine-tuning

Ximing Lu$^{♡♣}$   Faeze Brahman$^{♡♣}$   Peter West $^{♡♣}$   Jaehun Jung $^{♡}$
Khyathi Chandu $^{♣}$   Abhilasha Ravichander $^{♣}$   Lianhui Qin $^{♡}$
Prithviraj Ammanabrolu $^{♡♣}$   Liwei Jiang $^{♡♣}$   Sahana Ramnath $^{♢}$
Nouha Dziri $^{♣}$   Jillian Fisher $^{♡}$   Bill Yuchen Lin $^{♣}$   Skyler Hallinan $^{♡}$
Xiang Ren $^{♢♣}$   Sean Welleck $^{♡♣}$   Yejin Choi$^{♡♣}$

$^{♣}$Allen Institute for Artificial Intelligence
$^{♡}$University of Washington   $^{♢}$University of Southern California

## Abstract

While extreme-scale language models have demonstrated exceptional performance on a variety of language tasks, the degree of control over these language models through pure prompting can often be limited. Directly fine-tuning such language models can be effective for tailoring them, but it can be either extremely costly (e.g., GPT-3) or not even feasible for the broader community (e.g., GPT-4).

We propose **Inference-time Policy Adapters (IPA)**, which efficiently tailors a language model such as GPT-3 without fine-tuning it. IPA guides a large base model during decoding time through a lightweight policy adapter trained to optimize an arbitrary user objective with reinforcement learning.

On five challenging text generation tasks, such as toxicity reduction and lexically constrained generation, IPA consistently brings significant improvements over off-the-shelf language models. It outperforms competitive baseline methods, sometimes even including expensive fine-tuning. In particular, tailoring GPT-2 with IPA can outperform GPT-3, while tailoring GPT-3 with IPA brings a major performance boost over GPT-3 (and sometimes even over GPT-4). Our promising results highlight the potential of IPA as a lightweight alternative to tailoring extreme-scale language models.[1]

## 1   Introduction

Large language models (LLMs) have recently shown remarkable progress in various text generation tasks by adapting to instructions or examples (Ouyang et al., 2022; Brown et al., 2020). However, the degree of control (e.g., the inclusion of keywords, avoiding harmful language) offered by these extreme-scale models through pure prompting is still limited (Lou et al., 2023; Webson and Pavlick, 2021). Moreover, prompting can be

[1]Our code is publicly available at: https://github.com/GXimingLu/IPA

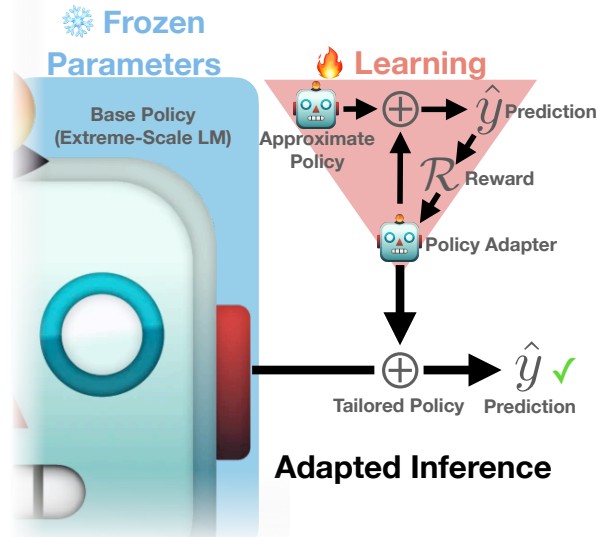

Figure 1: Inference-time Policy Adapters (IPA) efficiently steer a large-scale language model (such as GPT-3) during decoding-time through a lightweight policy adapter trained to optimize any arbitrary user objective with reinforcement learning.

a brittle process due to LLMs being overly sensitive to the surface-form of the instructions (Perez et al., 2021; Lu et al., 2022c). Furthermore, even with a carefully written prompt, LLMs may still struggle to fulfill certain task requirements due to their inherent limitations (Liu et al., 2022a; Zong and Krishnamachari, 2022).

Resource-intensive fine-tuning, through supervised learning, and more recently reinforcement learning (RL) (Lu et al., 2022a) have shown promise in tailoring language models to arbitrary user-given objectives. RL, in particular, known for its generalizability and flexibility, allows models to learn from desired rewards. However, these methods require accessing and updating models parameters, which can be extremely large or inaccessible in state-of-the-art models like GPT-4 (OpenAI, 2023b). This limitation makes fine-tuning unfeasible for the broader community.

Alternatively, *inference*-time algorithms can tailor a language model without accessing its parameters. These algorithms align language models' outputs with desired task/user-specific properties by adjusting the model's output distribution based on certain task-specific heuristics, while leaving the underlying model untouched. Despite the progress, these approaches are either restricted to specific tasks (Lu et al., 2021, 2020), require domain-specific knowledge (Liu et al., 2021a; Yang and Klein, 2021), suffer from expensive run-time at inference (Qin et al., 2022, 2021; Dathathri et al., 2020a), or have shown to be less effective compared to direct RL optimization (Lu et al., 2022a).

Drawing inspiration from RL and inference-time techniques, we propose **Inference-time Policy Adapters (🍺 IPA)**, an efficient and generalizable algorithm, which tailors a large language model at decoding-time toward desired objectives without fine-tuning it. To do so, IPA combines a large base LM's output distribution with that of a smaller-sized model (a lightweight **adapter policy**), and optimizes the combined distribution towards a given objective with RL (Figure 1). IPA uses two key ideas to make learning efficient. First, IPA *only updates the adapter's parameters*, avoiding the need to update the large base LM. Second, IPA replaces the large base model with an *approximate policy*–a smaller model that approximates the base model's distribution. The approximate policy is either a smaller model from the same language model family or a distilled version of the base model. At inference time, we decode with the combined distribution of the base model and the trained policy adapter.

Experiments across five challenging text generation tasks show that IPA brings consistent improvements over off-the-shelf language models, outperforming competitive baselines — sometimes even including expensive fine-tuning. In particular, tailoring GPT-2 with IPA can outperform GPT-3, while tailoring GPT-3 with IPA brings a major performance boost over GPT-3 (and sometimes even over GPT-4). Our compelling highlight the promise of IPA as a lightweight alternative for tailoring large language models to a wide range of objectives. IPA opens new ways to augment or customize extreme-scale language models using only academic-level resources.

## 2 Background

In this section, we introduce our text generation setting (§2.1) and a brief background on tailoring language models with reinforcement learning (§2.2). We then introduce our IPA algorithm for tailoring large language models without fine-tuning (§3).

### 2.1 Problem Setting

Text generation is the task of generating an output sequence $\mathbf{y}$ given an input sequence $\mathbf{x}$. We consider standard autoregressive language models, which decompose a sequence's probability as $p_\theta(\mathbf{y}|\mathbf{x}) = \prod_{t=1}^{|\mathbf{y}|} p_\theta(\mathbf{y}_t|\mathbf{y}_{<t}, \mathbf{x})$, where $p_\theta$ is a neural network with parameters $\theta$. Intuitively, our goal is to 'tailor' a pretrained model $p_\theta$ towards a user-specified objective (e.g., safety). Concretely, we assume that the objective is quantified by a reward function $\mathcal{R}(\mathbf{y}) \in \mathbb{R}$. We then aim to adjust $p_\theta$ so that its generated sequences have high reward and reasonable language quality (e.g., fluency).

### 2.2 Preliminary: Tailoring LMs with RL

Online policy-based reinforcement learning has emerged as an effective way to adjust a language model towards a reward function. Formally, these algorithms (e.g., PPO (Stiennon et al., 2022), Quark (Lu et al., 2022b), or NLPO (Ramamurthy* et al., 2023)) optimize a language model $p_\theta$ towards generating outputs $\mathbf{y}$ that maximize a given reward $\mathcal{R}$:

$$\theta^\star = \arg\max \mathbb{E}_{\mathbf{y} \sim p_\theta(\cdot|\mathbf{x})} \mathcal{R}(\mathbf{y}),$$

often along with regularization to maintain language quality. At a high-level, these algorithms use a policy $p_\theta$ to collect input-output examples, score the outputs with a reward function $\mathcal{R}$, and update parameter $\theta$ to maximize the expected reward. Although the exact optimization may differ, we can view any online policy-based RL algorithms as a functions $f_{\text{RL}}$ that take a policy $p_\theta$ and a reward function $\mathcal{R}$ as the inputs and outputs an optimized policy $p_{\theta^\star}$ with respect to $\mathcal{R}$. Formally,

$$f_{\text{RL}} : (p_\theta, \mathcal{R}; \theta') \to \theta^\star. \quad (1)$$

Here $\theta' \subseteq \theta$ denotes the subset of $p_\theta$'s parameters that are updated by the algorithm. The key idea behind IPA is to use a full model $p_\theta$ to collect examples, but update a small set of parameters $\theta'$.

## 3 Inference-time Policy Adapters (IPA)

We introduce Inference-time Policy Adapters (IPA), a lightweight approach to tailor language models

towards a user-specified objective. IPA trains a small *adapter policy* that adjusts the outputs of a (larger) base model at inference-time in order to maximize a reward. In doing so, IPA avoids the cost of updating the large base model, without the need to hand-design inference-time heuristics.

## 3.1 Policy Adaptation

We introduce the notion of '*tailoring*' used by IPA, which mainly involves three policies. First, IPA starts with a **base policy** $p_\theta$, which is the language model to tailor. Second, IPA introduces an **adapter policy** $p_\phi$, which is a language model with the same output space as the base policy (i.e., vocabulary), but different parameters $\phi$. Finally, IPA combines the base and adapter policies into a **tailored policy**:

**Definition 1** (Tailored policy). *The tailored policy $p_{\theta \leftarrow \phi}$ combines the distributions of the base policy $p_\theta$ and the adapter policy $p_\phi$,*

$$p_{\theta \leftarrow \phi}(\boldsymbol{y}_t | \boldsymbol{y}_{<t}) = \frac{1}{Z} p_\theta(\boldsymbol{y}_t | \boldsymbol{y}_{<t}) p_\phi(\boldsymbol{y}_t | \boldsymbol{y}_{<t}),$$

*where $Z$ is a normalization factor.*

The tailored policy is a product-of-experts (Hinton, 2002), which amounts to multiplying the next-token probabilities from the base and adapter policies, then normalizing the result. IPA's tailored policy has two key properties. First, it allows for adjusting the base policy's output without direct access to the base policy's parameters. This is critical for tailoring modern LLMs that provide access to the model's output distribution but not the model's parameters. Second, the policy adapter can use a much smaller model (i.e., $\phi \ll \theta$). This provides an efficient way to tailor a large base model.

## 3.2 Adapter Training with RL

Our goal is to adjust the tailored policy towards a user-specified objective. The key idea in IPA is to train the tailored policy to optimize a given reward with reinforcement learning, while *only updating the parameters of the adapter policy*.

Concretely, we use a reinforcement learning algorithm $f_{\text{RL}}$ (Eqn. 1) to optimize the tailored policy $p_{\theta \leftarrow \phi}$ with a reward function $\mathcal{R}$. Notably, we keep the base policy's parameters ($\theta$) frozen, and only update the adapter policy's parameters ($\phi$). That is,

$$\phi^\star = f_{\text{RL}}(p_{\theta \leftarrow \phi}, \mathcal{R}; \phi).$$

Intuitively, the adapter policy $p_\phi$ learns to rescale the frozen base policy $p_\theta$, yielding a tailored policy that is 'tailored to' the reward. Notice that our

framework does not depend on a specific RL algorithm, but rather treats RL as a flexible plug-in optimization tool. As we will demonstrate later, IPA proves to be effective when paired with three different RL algorithms (Lu et al., 2022b; Schulman et al., 2017; Ramamurthy et al., 2023), and in principle, it can easily integrate with others.

**Approximate Policy.** When the base model is extremely large (e.g., GPT-3), its forward pass is too costly to be used in the RL training loop. To overcome this, we propose using an **approximate policy** in IPA.

**Definition 2** (Approximate policy). *The approximate policy is defined as a smaller-sized neural model parameterized by $\hat{\theta}$ that approximates the distribution of the base policy and is used to replace the base policy in the RL-based adapter training:*

$$\phi^\star = f_{\text{RL}}(p_{\hat{\theta} \leftarrow \phi}, \mathcal{R}; \phi).$$

In practice, we can obtain an approximate policy in two different ways. First, we can use a *smaller pre-trained language model from the same model family*. We do this if the smaller model has similar conditional generation behavior as the base policy. For instance, we use an off-the-shelf GPT2-XL as the approximate policy to tailor GPT-3 in an open-ended generation. Alternatively, we can use a *distilled base policy* as the approximate policy. A distilled base policy is a language model trained on generations from the base policy, $\hat{\theta} = \arg\max \mathbb{E}_{\boldsymbol{y} \sim p_\theta(\cdot | \mathbf{x})} [\log P_{\hat{\theta}}(\boldsymbol{y})]$, known as sequence-level knowledge distillation (Kim and Rush, 2016; West et al., 2022). For example, to tailor GPT-3 for lexically constrained generation, we tune GPT2-XL on prompt-generation pairs from GPT-3 to get a distilled base policy.

**IPA at Inference Time.** At inference time, IPA uses the tailored policy $p_{\theta \leftarrow \phi}$ for decoding. Namely, at each time-step we obtain the next-token distribution from the tailored policy $p_{\theta \leftarrow \phi}(\mathbf{y}_t | \mathbf{y}_{<t})$, which can then be used with a standard decoding algorithm (e.g. nucleus sampling).

## 4 Experiments

We evaluate IPA on a diverse range of tasks: toxicity reduction (§4.1), lexically constrained generation (§4.2), open-ended generation (§4.3), dialogue safety control (§4.4), and knowledge-grounded dialogue (§4.5). In all benchmarks, IPA

consistently improve upon LLMs such as GPT-3 (text-davinci-003), surpassing competitive baselines and sometimes even outperforming expensive fine-tuned GPT-3 at a fraction of the cost.

## 4.1 Toxicity Reduction

LMs are susceptible to generating toxic completions, even when prompted with seemingly innocuous text (Gehman et al., 2020). Here, we assess IPA's efficacy in reducing toxicity from LMs.

**Datasets and Metrics.** The task is to generate a fluent continuation $y$ while avoiding offensive content for a given prompt $x$. We evaluate this on RE-ALTOXICITYPROMPTS benchmark (Gehman et al., 2020), which contains 100k prompts designed to elicit toxic generations. Following the experimental setup of Liu et al. (2021b), we use Perspective API to determine the average maximum toxicity across 25 sampled generations and the (empirical) toxicity probability of at least one toxic generation. In addition, we report fluency as the perplexity of generated output based on an off-the-shelf GPT2-XL model, and diversity as the count of unique n-grams normalized by the length of text. We also perform human evaluations; see Appendix A.1 for more details.

**Setup and Baselines** We apply IPA to tailor off-the-shelf GPT-2 and GPT-3[2]. To tailor GPT-2, we directly apply the base policy in the adapter training, denoted as IPA(GPT-2). For tailoring GPT-3, we use an off-the-shelf GPT-2 and a distilled GPT-3[3] as the approximate policy for the adapter training, labeled as IPA⁻(GPT-3) and IPA*(GPT-3) respectively. Notice that IPA⁻(GPT-3) is equivalent to directly applying the policy adapter trained to tailor GPT-2 on top of GPT-3. We initialize all the policy adapters with a pre-trained GPT2-L model.

We use QUARK as the RL algorithm in adapter optimization, and provide additional ablation studies to assess the effects of different RL algorithms. We use the Perspective API as the reward function, which provides a score ranging from 0 to 1 to indicate the degree of toxicity.

For tailoring GPT-2, we compare IPA with previously reported baselines from Lu et al. (2022a), including decoding-based methods: PPLM (Dathathri et al., 2020a), GeDi (Krause et al.,

[2]We refer text-davinci-003 as GPT-3 in this paper
[3]We finetune a GPT2-XL with prompt-output pairs from GPT-3 on REALTOXICITYPROMPTS as the distilled GPT-3.

| Models | Toxicity | | Fluency | Diversity | |
|---|---|---|---|---|---|
| | Avg Max. | Prob. | Pl. | Dist-2. | Dist-3. |
| *base policy:* GPT2-L | | | | | |
| GPT-2 | 0.527 | 0.520 | 11.31 | 0.85 | 0.85 |
| PPLM | 0.520 | 0.518 | 32.58 | 0.86 | 0.86 |
| GeDi | 0.363 | 0.217 | 60.03 | 0.84 | 0.83 |
| DEXPERTS | 0.314 | 0.128 | 32.41 | 0.84 | 0.84 |
| DAPT | 0.428 | 0.360 | 31.21 | 0.84 | 0.84 |
| PPO | 0.218 | 0.044 | 14.27 | 0.80 | 0.84 |
| QUARK | 0.196 | 0.035 | 12.47 | 0.80 | 0.84 |
| IPA (GPT-2) | **0.138** | **0.031** | **11.94** | 0.80 | 0.84 |
| *base policy:* GPT-3 | | | | | |
| GPT-3 | 0.275 | 0.197 | 10.65 | 0.78 | 0.81 |
| DEXPERTS | 0.223 | 0.112 | 23.41 | 0.79 | 0.82 |
| DAPT | 0.254 | 0.176 | 20.19 | 0.80 | 0.83 |
| IPA⁻ (GPT-3) | 0.150 | 0.056 | **10.34** | 0.79 | 0.81 |
| IPA* (GPT-3) | **0.101** | **0.028** | 12.68 | 0.79 | 0.83 |

Table 1: Automatic evaluation for *Toxicity Reduction* with off-the-shelf GPT2-large (top) and GPT-3 (bottom) as the base policy to tailor.

| RL Algo. | Toxicity | | Fluency | Diversity | |
|---|---|---|---|---|---|
| | Avg Max. | Prob. | Pl. | Dist-2. | Dist-3. |
| Quark | 0.138 | 0.031 | 11.94 | 0.80 | 0.84 |
| PPO | 0.125 | 0.029 | 12.47 | 0.80 | 0.84 |
| NLPO | 0.136 | 0.032 | 12.13 | 0.80 | 0.85 |

Table 2: Comparison of using different RL algorithm for training IPA for *Toxicity Reduction* with off-the-shelf GPT2-large as the base policy to tailor.

2021), DExpert (Liu et al., 2021a), and learning-based methods: DAPT (Gururangan et al., 2020), PPO (Schulman et al., 2017), and QUARK (Lu et al., 2022a). For tailoring GPT-3, we compare IPA to the baselines described above that are compatible with GPT-3's limited accessibility: DExpert (Liu et al., 2021a) and DAPT (Gururangan et al., 2020). We also provide runtime analysis in Appendix B.

**Results** As shown in Table 1, IPA outperforms all learning-based and decoding-based methods in tailoring GPT-2 and GPT-3, significantly reduces the toxicity while maintaining language quality. Interestingly, we found that applying the policy adapter optimized for GPT-2 directly on top of GPT-3 (i.e., IPA⁻) is highly effective, showcasing the adaptability and reusability of IPA. Notably, when tailoring GPT-3, IPA outperforms the costly domain adaptive training (DAPT), which exhaustively fine-tune GPT-3 on a non-toxic corpus. This further emphasizes the promise of the IPA as a cost-efficient approach to align LLMs. Our findings are further confirmed by human evaluation (Appendix A.1).

Finally, we conduct ablations on the effect of

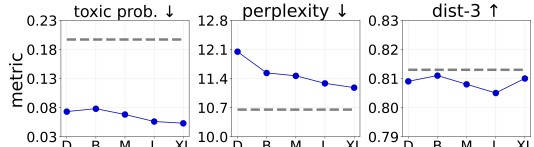

Figure 2: Performance of IPA⁻ (**blue line**) with respect to the size of the adapter model (distill-GPT2, GPT2-small, GPT2-medium, GPT2-large, GPT2-XL) on top of a off-the-shelf GPT-3 as the base policy. The **grey line** denotes the performance of the off-the-shelf GPT-3.

| Models | Automatic | | Human | | |
|---|---|---|---|---|---|
| | Cov. | Fl. | Qu. | Pl. | Overall |
| GPT-3 | 37.01 | 94.89 | 2.84 | 2.81 | 2.60 |
| GPT-3.5 | 65.17 | **95.89** | 2.93 | 2.88 | 2.90 |
| GPT-4 | 84.81 | 95.49 | **2.95** | **2.97** | **2.96** |
| GPT-3$_{sft}$ | 72.89 | 73.96 | 2.56 | 2.60 | 2.50 |
| IPA* (GPT-3) | **88.54** | 92.58 | 2.90 | 2.87 | 2.88 |

Table 3: Automatic and human evaluation results for *Lexically Constrained Generation*. Human evaluation scores are on a 3-point Likert Scale.[4]

RL algorithms. As shown in Table 2, IPA is effective with various RL algorithms, all of which lead to state-of-the-art performance. Additional ablation in Figure 2 shows that a policy adapter as small as a distilled GPT-2 can effectively tailor the ×1000 larger GPT-3 model, achieving comparable performance with our main result.

## 4.2 Lexically Constrained Generation

Next, we test IPA in lexically constrained generation. We consider a more challenging setup of *ordered lexical constraints*, where the generation is considered correct if it includes all the keywords with the correct order specified in the input prompt.

**Datasets and Metrics.** We use COMMONGEN (Lin et al., 2020), a dataset for generative commonsense reasoning. We deliberately instruct the models to generate a sentence with the given keywords while following the order they appear in the input prompt. For automatic evaluation, we gauge the constraint satisfaction with *coverage*, a binary metric that evaluates a generation to be correct only when it includes all the keywords and also matches the specified order. We also measure the *fluency* using a critic model fine-tuned on CoLA (Warstadt et al., 2019). For human evaluation, we assess the *quality* and *plausibility* of model generations for 100 randomly sampled test examples based on a 3-point Likert Scale; see details in Appendix E.

**Setup and Baselines.** As we will demonstrate later, zero-shot GPT-3 is surprisingly poor at satisfying ordered lexical constraints, even with explicit instructions. Our goal is to make GPT-3 more reliable in constraint satisfaction. We use distilled GPT3 [5] as the approximate policy for adapter training, since an off-the-shelf GPT-2 cannot perform constrained generation out of the box. We initialize the policy adapter with a pre-trained GPT2-L model. We use QUARK as the RL algorithm and choose our reward to be the product of the coverage score and the fluency score, as this promotes constraint satisfaction and fluency preservation. Please see Appendix A.4 for more reward analysis.

We compare IPA with its base policy GPT-3, as well as more advanced LLMs: GPT-3.5 and GPT-4 (OpenAI, 2023a). As a strong supervised baseline, we also fine-tune GPT-3 on the COMMONGEN train set, which contains human-written outputs with the correct lexical order, denoted as GPT-3$_{sft}$.

**Results.** As shown in Table 3, powerful LMs such as GPT-3 often struggle to satisfy ordered lexical constraints even with explicit instructions. IPA leads to remarkable improvement on top of GPT-3 and surpasses more advanced models such as GPT-3.5 and GPT-4 in terms of constraint coverage, while achieving better or comparable generation quality. Noticeably, IPA outperforms fine-tuned GPT-3 in both constraint coverage and generation quality at a fraction of its cost: while fine-tuning GPT-3 costs $156.82, training a distilled GPT-3 as the approximate policy requires only $28.59 for generating outputs from GPT-3. Our results highlight the potential of the IPA as a cost-efficient way to enhance the capabilities of LLMs.

## 4.3 Open-ended generation

We further evaluate IPA on open-ended generation, following the experimental setup in (Li et al., 2022b). The goal is to make machine-generated content more fluent, coherent, and human-like.

**Datasets and Metrics.** We experiment on the news domain using XSum dataset (Narayan et al., 2018). Following Li et al. (2022b), we use the first 32 words as our input prompt, and generate 84 tokens as continuations. We evaluate using both

---

[4]Human pairwise agreements are 0.97, 0.94, and 0.93 for quality, plausibility and overall, respectively.

[5]We finetune a GPT2-XL with prompt-output pairs from GPT-3 on COMMONGEN train set as the distilled GPT-3

| Decoding Method | Diversity | Coherence | Critic | Mauve |
|---|---|---|---|---|
| *base policy*: GPT2-XL | | | | |
| greedy | 55.05 | 49.57 | 7.88 | 15.32 |
| top-k (k=50) | 92.60 | 48.53 | 10.72 | 53.13 |
| top-p (p=0.95) | 95.85 | 47.61 | 13.24 | 56.42 |
| typical ($\tau$=0.95) | 95.80 | 46.08 | 23.49 | 63.92 |
| SimCTG | 95.67 | 46.12 | 23.67 | 62.21 |
| Contrastive | 95.99 | 49.42 | 36.73 | 61.95 |
| IPA (GPT2-XL) | **96.12** | **51.81** | **50.93** | **84.18** |
| *base policy*: GPT-3 | | | | |
| top-p (p=0.95) | 95.63 | 56.16 | 18.58 | 63.73 |
| IPA⁻ (GPT-3) | 95.35 | 57.26 | 22.62 | 71.40 |
| IPA* (GPT-3) | **96.26** | **61.94** | 32.84 | **73.17** |

Table 4: Automatic evaluation for *open-domain generations* on XSum with off-the-shelf GPT2-XL (top) and GPT-3 (bottom) as the base policy to tailor. Critic scores refer to *human-likeness* according to OpenAI detector.

automatic and pairwise human evaluation. For automatic evaluation, we use aggregate n-gram diversity and coherence scores (Li et al., 2022b) as well as MAUVE (Pillutla et al., 2021), which measures the distribution similarity between the set of human-written and machine-generated texts. To measure the *human-likeness* of generated texts, we employ OpenAI detector[6], a classifier for distinguishing AI vs. human-written text. We use the classifier's probability assigned to 'human' text to serve as an additional metric, denoted as Critic. For human evaluation, we randomly sample 100 test examples and perform pairwise comparisons of our method against baselines on *coherence* and *fluency* using AMT; see details in Appendix E.

**Setup and Baselines.** We apply IPA to tailor off-the-shelf GPT2-XL and GPT-3, following the same setup as toxicity reduction task (section 4.1). Same as before, the tailor policies are denoted as IPA(GPT-2), IPA⁻(GPT-3) and IPA*(GPT-3), respectively. We use QUARK as the RL algorithm and the product of diversity, coherence, and critic scores as the reward function. We found it critical to combine multiple metrics as the reward function to improve the overall generation quality; see Appendix A.4 for more analysis on reward functions. For tailoring GPT-2, we compare decoding with IPA with six different decoding strategies: greedy, top-k sampling ($k = 50$), nucleus sampling ($p = 0.95$), typical sampling ($\tau = 0.95$) (Meister et al., 2023), SimCTG (Su et al., 2022), and Contrastive decoding (Li et al., 2022b). The latter three are

[6]https://github.com/promptslab/openai-detector

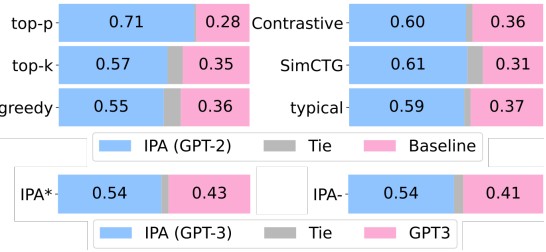

Figure 3: Pairwise human evaluation in terms of **overall quality** for *Open-ended Generation* on XSum with off-the-shelf GPT2-XL (top) and GPT-3 (bottom) as the base policy to tailor.[7]

| Models | Automatic | Human | |
|---|---|---|---|
| | Safety | Safety | Coherence |
| DialoGPT | 0.46 | 1.34 | 2.45 |
| Godel | 0.49 | 1.40 | 2.53 |
| Blenderbot | 0.53 | 1.43 | 2.60 |
| ChatGPT | 0.74 | **1.60** | 2.68 |
| IPA⁻ (BlenderBot-3B) | **0.78** | 1.57 | **2.75** |

Table 5: Automatic and human evaluation results for *Dialogue Safety Control*. Human evaluation scores are on a 3-point Likert Scale.[8]

specifically designed to improve the coherence and naturalness of the generated text. For tailoring GPT-3, we compare IPA with GPT-3's default generation technique: decoding with nucleus sampling ($p = 0.95$). as other decoding methods are not applicable to GPT-3 due to its limited API access.

**Results.** As shown in Table 4, IPA significantly outperforms all previous baselines in tailoring GPT-2 and GPT-3 across all automatic metrics. Notably, it achieves an absolute improvement of 20.26% over the best-performing baseline in the Mauve score. Our pairwise human evaluation in Figure 3 also verify the results. IPA generates significantly more coherent and fluent texts compared to other baselines. Overall, on average, human evaluators preferred IPA 1.8× more than other baselines. Interestingly, we found that directly applying the policy adapter optimized for GPT-2 on top of GPT-3 (i.e., IPA⁻) significantly improves the generation quality, highlighting the adaptability and reusability of IPA. We observed further improvement when using distilled GPT-3 as the approximate policy (i.e., IPA*). Our promising results once again showcase the effectiveness and efficiency of IPA.

## 4.4 Dialogue Safety Control

Existing dialogue systems often fail to respond safely to potentially unsafe user utterances (Kim et al., 2022), limiting their deployment in real-world applications. Here, we aim to evaluate IPA for controlling the safety of a dialogue model.

**Datasets and Metrics.** We experiment on DI-ASAFETY (Sun et al., 2022), a challenging dataset containing 54K context-sensitive unsafe examples. The task is to generate a coherent response to a potentially unsafe utterance while avoiding offensive, harmful, toxic or biased language. DIASAFETY contains human-written safe and unsafe responses which we use to train a dialogue safety classifier. We use the classifier score as an automatic measure of safety. In addition, we conduct a human evaluation of *safety* and *coherence* (3-point Likert scale) on 200 examples through Amazon Mechanical Turk; see Appendix E Figure 4 for details.

**Setup and Baselines.** We apply IPA to tailor the Blenderbot family models (Roller et al., 2021), which are pretrained dialogue agents. Specifically, we use Blenderbot-3B-distill as the frozen base policy, a samller Blenderbot-1B-distill as the approximate policy and initialize the policy adapter with a Blenderbot-1B-distill model. We use QUARK as the RL algorithm for adapter training. To preserve the dialogue quality while controlling the response safety, we choose our reward to be the product of the safety score from our dialogue safety classifier, as well as coherence and engagingness scores from UniEval-Dialogue (Zhong et al., 2022).[9]

We compare IPA with its base policy, i.e., Blenderbot-3B-distill, and other off-the-shelf dialogue models including DialoGPT (Zhang et al., 2020), GODEL (Peng et al., 2022) as well as Chat-GPT (OpenAI, 2022). ChatGPT is known to have safeguards through content filtering and is considered a strong baseline.

**Results.** As shown in Table 5, IPA significantly improves dialogue safety and coherence compared to its base policy Blenderbot-3B-distill, surpassing other dialogue models including DialoGPT and GODEL. In comparison with ChatGPT, IPA achieves comparable performance on safety based

on both automatic and human evaluation while showcasing improved coherence. Upon further investigation, we found that ChatGPT often generates canned responses like "I'm a language model; I'm not allowed..." as hard safeguards, which hurts the coherence and naturalness of the dialogue flow. On the other hand, Blenderbot tailored by IPA can generate safe responses that are coherent, natural, and human-like. Our results demonstrate the potential of IPA to enhance controllability in various NLP applications beyond conditional text generation.

## 4.5 Knowledge-grounded Dialogue

Ideally, knowledge-grounded dialogue systems should generate responses that are faithful to the given knowledge. However, models tend to generate hallucination containing unverifiable information (Dziri et al., 2022a; Rashkin et al., 2021a; Dziri et al., 2022c). To address this undesirable behavior, we use IPA to tailor dialogue model towards generating more faithful content. Given the knowledge $K$ and the conversation history $H$, the task is to generate a response $r$ that's faithful to $K$ and coherent with $H$.

**Dataset and Metrics** We evaluate on the Wizard of Wikipedia (WoW) data. WoW (Dinan et al.) involves a Wizard and an Apprentice engaging in a conversation. The Wizard's role is to provide information on a specific topic, while the Apprentice's task is to seek further details. WoW has been shown to suffer from hallucinations (Dziri et al., 2022b), in more than 60% of the turns, making it a valuable dataset for studying hallucination issues. FaithDial (Dziri et al., 2022a) is a hallucination-free benchmark created by modifying the hallucinated responses within the WoW dataset. We use the FaithDial test data at test time to evaluate the faithfulness of responses and compare them against the knowledge snippets and gold responses.

To measure faithfulness, we use the critic model (Dziri et al., 2022a), which returns the probability of an given utterance being identified as faithful. Additionally, we use BERTScore to measure the semantic similarity between the generated response $r$ and the knowledge $K$, and the token-level F1 score to rate the lexical overlap between $r$ and $K$. To measure coherence and engagingness, we use the UniEval model (Zhong et al., 2022).

**Setup and Baselines** Similar to the dialogue safety experiment, we use the Blenderbot-{3, 1}B-distill model (Roller et al., 2021) as our base policy

---

[7]Average pairwise agreements are 0.88 and 0.82 with GPT2-XL and GPT-3, respectively.

[8]Human pairwise agreements are 0.84 and 0.87 for safety and coherence.

[9]https://github.com/maszhongming/UniEval

| Dialogue Model | Critic | BERTScore | F1 | Coherence | Engaging |
|---|---|---|---|---|---|
| *supervised* baseline | | | | | |
| GPT-2 | 39.9 | 0.29 | 47.7 | 0.77 | 1.26 |
| DIALOGPT | 40.6 | 0.34 | 53.5 | 0.83 | 1.32 |
| DOHA | 46.8 | 0.32 | 56.1 | 0.88 | 1.33 |
| T5 | 53.5 | 0.41 | 61.7 | 0.86 | 1.28 |
| T5-CTRL | 54.8 | 0.45 | 65.2 | 0.83 | 1.21 |
| T5-LT | 58.6 | 0.43 | 65.0 | 0.83 | 1.21 |
| *off-the-shelf* dialogue model | | | | | |
| BlenderBot | 10.3 | 0.12 | 9.8 | **0.92** | 1.21 |
| IPA⁻ (BlenderBot) | **76.6** | **0.68** | **80.1** | 0.91 | **1.34** |

Table 6: Evaluation results for *Knowledge-Grouded Dialogue* generations on Faithdial. We use off-the-shelf Blenderbot as the base policy to tailor.

and approximate policy respectively, and initialize the policy adapter with a Blenderbot-1B-distill model. We use QUARK as the RL algorithm. To preserve coherence and engagingness while ensuring the faithfulness of a dialogue response, we choose our reward to be the product of the faithfulness score from the critic model described above, as well as coherence and engagingness scores from UniEval-Dialogue (Zhong et al., 2022).

We compare to previously baselines from Dziri et al. (2022a), supervised models fine-tuned on WoW, including GPT2, DialoGPT (Zhang et al., 2020), DoHA (Prabhumoye et al., 2021) T5 (Raffel et al., 2020), T5-CTRL (Rashkin et al., 2021b), and T5-LossTruncation (Kang and Hashimoto, 2020). We also compare against the base policy, off-the-shelf BlenderBot model (Roller et al., 2021).

**Results**    As shown in Table 6, supervised models struggle to generate faithful dialogue response grounded on the given knowledge. This is mainly because of the poor data quality of their supervision dataset: WoW has been shown to suffer from hallucinations in more than 60% of the turns (Dziri et al., 2022a). Moreover, pre-trained dialogue models like Blenderbot demonstrate even worse performance at generating faithful response, despite being trained on WoW and other knowledge-grounded dialogue datasets in their pre-training stage. IPA significantly improves the faithfulness of the generated dialogue response over its base policy Blenderbot while preserving the dialogue quality (i.e., coherence and engagingness), outperforming all other baselines. Our results showcases the potential of IPA to improve reliability and trustworthiness in various downstream applications.

## 5   Related Work

**Controlled Decoding**    Recent studies have explored controlled generation at inference time by designing new decoding algorithms (Keskar et al., 2019; Mireshghallah et al., 2022; Li et al., 2022a; Chen et al., 2022; Zhang et al., 2022). For example, Neurologic decoding (Lu et al., 2020), and GBS (Hokamp and Liu, 2017) generalize beam search for lexically constrained decoding, by constraining decoding space with keyword-related penalties. DExperts (Liu et al., 2021b) modifies output distribution during decoding with attribute-specific expert models. Another line of research develops gradient-based decoding for more general control (Qin et al., 2020, 2022; Sha, 2020; Dathathri et al., 2020b; Kumar et al., 2021). For example, COLD Decoding (Qin et al., 2022) introduces energy-based modeling to impose arbitrary constraints on text and samples with Langevin dynamics. Despite their progress, these approaches either are designed for particular control types or rely on computationally expensive gradient computations.

**Reinforcement Learning for NLG**    RL has historically been used in multiple NLG tasks such as machine translation (Wu et al., 2016; Nguyen et al., 2017), summarization (Paulus et al., 2017), dialogue (Li et al., 2016; Zhou et al., 2017), text games (Narasimhan et al., 2015; Hausknecht et al., 2020), etc to optimize for an arbitrary non-differentiable reward. This was often done using online policy gradient methods such as RE-INFORCE (Sutton and Barto, 2018), leading to documented issues with reward hacking (Choshen et al., 2020; Kiegeland and Kreutzer, 2021). Recent advances introduce a KL reward penalty which significantly increases the naturalness of generated text (Ouyang et al., 2022; Korbak et al., 2022). This method has been used extensively to tune a base LM via online on-policy (Ramamurthy* et al., 2023), off-policy (Guo et al., 2022; Lu et al., 2022b), and offline (Snell et al., 2023; Korbak et al., 2023) RL. Such methods quickly become computationally infeasible for extreme-scale LMs.

## 6   Conclusion

we present IPA, a lightweight inference-time policy adapter that tailor a frozen large language model towards desirable properties (e.g., safety, coherence) in an efficient, generalizable, and flexible way.    Specifically, IPA combines the generaliz-

ability of RL with the plug-and-play flexibility of inference-time techniques, permitting customization of large language models without the need for costly fine-tuning. Extensive experiments across five challenging text generation tasks show that IPA brings consistent improvements over LLMs, outperforming competitive baselines — sometimes even surpassing expensive fine-tuning. We hope our work sheds light on creative and efficient algorithmic innovations to complement the pursuit of model scales with academic-level resources.

## 7 Limitations and Ethical Consideration

While the versatility of the IPA is a crucial feature that enables aligning large language models with arbitrary user-given objectives, it may also pose potential dual-use concerns, especially when combined with the power of large language models.

First, as with any controllable text generation technique, IPA could be potentially used for unintended malicious purposes, such as manipulating models to produce hateful, toxic content or misinformation. As malicious users can already exploit any existing techniques for harmful purposes theoretically, we foresee minimal risk introduced by IPA specifically. Nevertheless, we highly recommend avoiding such negative applications of IPA.

Moreover, similar to any RL-based method that depends on the reward function for learning signals, IPA is susceptible to the innate shortcomings from the reward model. For instance, we use the Perspective API calls as the reward function for the toxicity reduction task; any limitations or potential biases from these public API calls will propagate into the learning of IPA. Nonetheless, as more accurate, transparent, and inclusive classifiers are developed, we anticipate that IPA would inherit those improvements as well.

Beyond these two primary concerns, another inherent limitation of IPA is its requirement to access the output logits of the base LM. This constraint hinders IPA's compatibility with certain models, such as GPT-4, which permit access only to the output, not the logits. Finally, like general RL frameworks, IPA relies on the assumption that user objectives are quantifiable through a reward function. However, this premise may not always hold, particularly when user objectives are inherently challenging to measure, thus limiting IPA's applicability.

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

| | IPA- vs. GPT3 | | IPA- vs. DEXPERTS | | IPA- vs. DAPT | |
|---|---|---|---|---|---|---|
| **Less Toxic** | **0.17** | 0.09 | **0.15** | 0.09 | **0.13** | 0.12 |
| **More Topical** | 0.20 | **0.21** | **0.23** | 0.14 | **0.22** | 0.20 |
| **More Fluent** | **0.27** | 0.23 | **0.24** | 0.16 | **0.21** | 0.18 |

| | IPA* vs. GPT3 | | IPA* vs. DEXPERTS | | IPA* vs. DAPT | |
|---|---|---|---|---|---|---|
| **Less Toxic** | **0.18** | 0.05 | **0.14** | 0.06 | **0.15** | 0.10 |
| **More Topical** | 0.23 | 0.23 | **0.28** | 0.17 | 0.18 | 0.18 |
| **More Fluent** | **0.26** | 0.21 | **0.32** | 0.15 | **0.23** | 0.22 |

Table 7: Human evaluation results of *Toxicity Reduction*, comparing the percentage of texts rated as less toxic, more topical, and more fluent as generated by IPA- and IPA* versus other baselines.

| Models | Toxicity | | Fluency | Diversity | |
|---|---|---|---|---|---|
| | Avg Max. | Prob. | Pl. | Dist-2. | Dist-3. |
| GPT-3 (zero-shot) | 0.275 | 0.197 | 10.65 | 0.78 | 0.81 |
| GPT-3 (5-shot) | 0.214 | 0.132 | 15.96 | 0.76 | 0.80 |
| GPT-3 (10-shot) | 0.208 | 0.145 | 17.83 | 0.77 | 0.80 |
| IPA- (GPT3) | 0.150 | 0.056 | **10.34** | 0.79 | 0.81 |
| IPA* (GPT3) | **0.101** | **0.028** | 12.68 | 0.79 | 0.83 |

Table 8: Automatic evaluation results for *Toxicity Reduction* with off-the-shelf GPT-3.

## A    Further Experiment

### A.1    Human Evaluation for Toxicity

We perform additional pairwise human evaluation on tailoring GPT-3 to reduce toxicity. We compare the outputs from IPA* and IPA- to each baseline, based on the perceived level of toxicity (which one is less rude or disrespectful), topicality (which one is more natural, relevant, and logical), and fluency (which one is more grammatically correct and coherent), on 100 random prompts from the test set of REALTOXICITYPROMPTS using.

As shown in Table 7, the human evaluation results confirms that both IPA- and IPA* effectively tailor GPT-3 to be less toxic while maintaining the language quality. This again underscores the potential of IPA as a cost-effective method for aligning large language models with user-defined objectives.

### A.2    Additional Baseline: Few-shot

In the experimental section, we show that in zero-shot setting LLMs such as GPT-3 often struggle to fulfill users' requests, such as generating safe content or reliably satisfying lexical constraints. Here, we conduct additional experiment to access LM's performance in few-shot setting on toxicity reduction and lexically constrained generation.

As illustrated in Table 8 and Table 9, prompting GPT-3 with additional few-shot examples improves its performance to some extent, but it still falls short of consistently fulfill users' requests. The gain is

| Models | Coverage | Fluency |
|---|---|---|
| GPT-3 (zero-shot) | 37.01 | **94.89** |
| GPT-3 (5-shot) | 43.85 | 94.34 |
| GPT-3 (10-shot) | 45.70 | 94.21 |
| IPA* (GPT-3) | **88.54** | 92.58 |

Table 9: Automatic evaluation results for *Lexically Constrained Generation* with off-the-shelf GPT-3.

| Models | Coverage | Fluency |
|---|---|---|
| LLaMA | 28.73 | 89.64 |
| IPA- (LLaMA) | **81.49** | **89.71** |

Table 10: Automatic evaluation results for *Lexically Constrained Generation* with off-the-shelf LLaMA-13B as the base policy to tailor.

particularly limited in lexically constrained generation, likely due to GPT-3's inherent limitations when dealing with hard logical constraints. Importantly, IPA on top of zero-shot GPT-3 outperforms all the few-shot baselines by a noticeable margin across all scenarios. The results further highlight the importance of our method, which directly optimize the base policy to align with user-specified objectives instead of solely relying on the innate capabilities of LLMs through prompting.

### A.3    Additional Experiments with LLaMA

We conducted additional experiments with LLaMA models (Touvron et al., 2023) for the constrained generation task. We apply IPA to tailor an off-the-shelf LLaMA-13B model and initialize the policy adapter with a LLaMA-7B model. As shown in Table 10, IPA leads to remarkable improvement on top of LLaMA-13B in terms of constraint coverage while maintaining language quality.

### A.4    Reward Analysis

We provide further analysis to justify our selection of reward functions for each task.

**Toxicity Reduction**    Following previous work Lu et al. (2022b), we use the Perspective API score as a reward function, which provides a score between 1 (non-toxic) and 0 (toxic). We observe that IPA effectively reduce the toxicity while preserving the language quality in terms of fluency and diversity in both automatic and human evaluation.

**Lexically Constrained Generation**    Our goal is to enhance constraint satisfaction. As shown in Table 11, optimizing for constraint coverage alone may result in a slight decline in language fluency,

| Reward | Coverage | Fluency |
|---|---|---|
| coverage | **90.75** | 83.91 |
| coverage, fluency | 88.54 | **92.58** |

Table 11: Automatic evaluation results for *Lexically Constrained Generation* with off-the-shelf GPT-3 as the base policy using different reward functions

| Reward | Diversity | Coherence | Critic | Mauve |
|---|---|---|---|---|
| coherence | 92.41 | **64.98** | 5.41 | 68.25 |
| coherence, critic | 93.73 | 51.03 | **52.36** | **84.32** |
| coherence, critic, diversity | **96.12** | 51.81 | 50.93 | 84.18 |

Table 12: Automatic evaluation for *open-domain generations* on XSum with off-the-shelf GPT2-XL as the base policy using different reward functions.

as measured by COLA. However, by incorporating fluency as an auxiliary reward, we notice improvements in both dimensions. Human evaluations further support our findings.

**Open-ended Generation**  The goal is to make machine-generated content more fluent, coherent, and human-like. As shown in Table 12, optimizing solely for coherence does not yield significant improvements in the overall generation quality, as evaluated by MAUVE. Incorporating scores from the OpenAI detector, a classifier for distinguishing between AI vs. human-written text, as an additional reward serves as an essential element in improving the overall quality and human-likeness of generated texts. Moreover, we found that integrating diversity score as another auxiliary reward helps maintain the diversity of generations while promoting higher quality output.

**Dialogue Safety Control**  Our aim to improving the safety of a dialogue model. As shown in Table 13, optimizing for safety score alone may result in a decrease in the overall quality of the generated dialogue, measured by coherence, engagingness and overall score from UniEval-Dialogue (Zhong et al., 2022). The generated responses tends to be bland and templated, such as "I don't know...", "I'm not sure...". We found that integrating coherence and engagingness scores as additional reward helps preserving natural dialogue flow while promoting safe responses.

**Knowledge-grounded Dialogue**  Our aim to improving the faithfulness of dialogue response with respect to the given knowledge. As shown in Table 14, optimizing for faithfulness score alone may result in a decrease in the overall quality of the

| Reward | Safety | Coherence | Engaging | Overall |
|---|---|---|---|---|
| safety | **0.85** | 0.82 | 1.32 | 0.88 |
| safety, coherence, engaging | 0.78 | **0.90** | **1.91** | **0.98** |

Table 13: Evaluation results for *Dialogue Safety Control* on DIASAFETY with different reward functions.

| Reward | Critic | Coherence | Engaging | Overall |
|---|---|---|---|---|
| critic | **85.3** | 0.84 | 1.01 | 0.88 |
| critic, coherence, engaging | 76.6 | **0.91** | **1.34** | **0.97** |

Table 14: Evaluation results for *Knowledge-Grounded Dialogue* on Faithdial with different reward functions.

generated dialogue, measured by coherence, engagingness and overall score from UniEval-Dialogue (Zhong et al., 2022). The generated responses are often the exact copy of the given knowledge, lacking of abstractiveness. We found that integrating coherence and engagingness scores as additional reward helps preserving the naturalness of the generated responses while enhancing their faithfulness.

# B  Runtime Analysis

We conduction additional runtime analysis on toxicity reduction task, comparing the inference speed of IPA with other baseline methods. As shown in Table B, IPA is significantly more efficient than most of the baseline methods and falls within a similar range as nucleus sampling.

| Method | Runtime |
|---|---|
| Nucleus Sampling | 0.03 |
| PPLM (Dathathri et al., 2020a) | 23.7 |
| GeDi (Krause et al., 2021) | 0.78 |
| Dexperts (Liu et al., 2021a) | 0.12 |
| DAPT (Gururangan et al., 2020) | 0.03 |
| QUARK (Lu et al., 2022a) | 0.03 |
| Inference-time Policy adapter | 0.08 |

Table 15: Inference runtime (seconds per sentence generation) of IPA versus other baseline methods with GPT2-L as the base policy on toxicity reduction task.

# C  Experiment Detail

## C.1  Off-the-Shelf Models

We download off-the-shelf models, including pretrained GPT-2 and BlenderBot, from HuggingFace Transformers (Wolf et al., 2020), which are implemented in the PyTorch deep learning framework. We access GPT-3, GPT-3.5 and GPT-4 models via API calls through OpenAI platform.

## C.2 Model Training Details

All training is performed on 8 NVIDIA Quadro RTX 8000 GPUs and costs about 3000 GPU hours in total. Our method is implemented with PyTorch an the Huggingface Transformers library.

### C.2.1 Toxicity Reduction

We initialize the policy adapter with an off-the-shelf GPT2-L model and use QUARK as the RL algorithm for the adapter training. Hyperparameters for training are given in Table 16. We performed a hyperparameter grid search for the number of training steps over the range [10k, 20k], for the KL coefficient $\beta$ over the range [0, 0.3], and for the frequency of exploration over the range [5, 20]. During inference, we use nucleus sampling with $p = 0.9$ and temperature 1.0.

| Hyperparameter | Assignment |
|---|---|
| model | GPT2-Large |
| number of parameters | 774M |
| number of steps | 18000 |
| batch size | 64 |
| learning rate optimizer | Adam |
| Adam epsilon | 1e-8 |
| Adam initial learning rate | 1e-5 |
| learning rate scheduler | linear with warmup |
| warmup steps | 800 |
| KL coefficient $\beta$ | 0.05 |
| frequency of exploration | 8 |

Table 16: Hyperparameters for training policy adapter to reduce toxicity

### C.2.2 Lexically Constrained Generation

We initialize the policy adapter with an off-the-shelf GPT2-L model and use QUARK as the RL algorithm for the adapter training. Hyperparameters for training are given in Table 17. We performed a hyperparameter grid search for the number of training steps over the range [5k, 20k], for the KL coefficient $\beta$ over the range [0, 0.3], and for the frequency of exploration over the range [10, 30]. During inference, we use nucleus sampling with $p = 0.9$ and temperature 1.0.

### C.2.3 Open-ended generation

We initialize the policy adapter with an off-the-shelf GPT2-L model and use QUARK as the RL algorithm for the adapter training. Hyperparameters for training are given in Table 18. We performed a hyperparameter grid search for the number of training steps over the range [30k, 50k], for the KL coefficient $\beta$ over the range [0, 0.3], and for the

| Hyperparameter | Assignment |
|---|---|
| model | GPT2-Large |
| number of parameters | 774M |
| number of steps | 14000 |
| batch size | 64 |
| learning rate optimizer | Adam |
| Adam epsilon | 1e-8 |
| Adam initial learning rate | 1e-5 |
| learning rate scheduler | linear with warmup |
| warmup steps | 500 |
| KL coefficient $\beta$ | 0.01 |
| frequency of exploration | 15 |

Table 17: Hyperparameters for training policy adapter to lexically constrained generation

frequency of exploration over the range [15, 25]. During inference, we use nucleus sampling with $p = 0.9$ and temperature 1.0.

| Hyperparameter | Assignment |
|---|---|
| model | GPT2-Large |
| number of parameters | 774M |
| number of steps | 50000 |
| batch size | 64 |
| learning rate optimizer | Adam |
| Adam epsilon | 1e-8 |
| Adam initial learning rate | 1e-5 |
| learning rate scheduler | linear with warmup |
| warmup steps | 1500 |
| KL coefficient $\beta$ | 0.05 |
| frequency of exploration | 25 |

Table 18: Hyperparameters for training policy adapter to open-ended generation

### C.2.4 Dialogue Safety Control

We initialize the policy adapter with an off-the-shelf blenderbot-1B-distill model and use QUARK as the RL algorithm for the adapter training. Hyperparameters for training are given in Table 19. We performed a hyperparameter grid search for the number of training steps over the range [10k, 15k], for the KL coefficient $\beta$ over the range [0, 0.3], and for the frequency of exploration over the range [10, 30]. During inference, we use nucleus sampling with $p = 0.6$ and temperature 1.0.

### C.2.5 Knowledge-grounded Dialogue

We initialize the policy adapter with an off-the-shelf blenderbot-1B-distill model and use QUARK as the RL algorithm for the adapter training. Hyperparameters for training are given in Table 20. We performed a hyperparameter grid search for the number of training steps over the range [7.5k, 15k], for the KL coefficient $\beta$ over the range [0, 0.3], and

| Hyperparameter | Assignment |
|---|---|
| model | blenderbot-1B-distill |
| number of parameters | 1B |
| number of steps | 15000 |
| batch size | 64 |
| learning rate optimizer | Adam |
| Adam epsilon | 1e-8 |
| Adam initial learning rate | 1e-5 |
| learning rate scheduler | linear with warmup |
| warmup steps | 300 |
| KL coefficient $\beta$ | 0.1 |
| frequency of exploration | 15 |

Table 19: Hyperparameters for training policy adapter to control dialogue safety

for the frequency of exploration over the range [15, 30]. During inference, we use nucleus sampling with $p = 0.6$ and temperature 1.0.

| Hyperparameter | Assignment |
|---|---|
| model | blenderbot-1B-distill |
| number of parameters | 1B |
| number of steps | 12500 |
| batch size | 64 |
| learning rate optimizer | Adam |
| Adam epsilon | 1e-8 |
| Adam initial learning rate | 1e-5 |
| learning rate scheduler | linear with warmup |
| warmup steps | 300 |
| KL coefficient $\beta$ | 0.1 |
| frequency of exploration | 25 |

Table 20: Hyperparameters for training policy adapter to improve dialogue faithfulness

## D   Additional Related Works

**Parameter-Efficient Fine-Tuning**  Prompting and prefix-tuning (Li and Liang, 2021) adapt a very large model to a specific task. However, they are affected by sensitivity based on order of words or examples (Zhao et al., 2021; Webson and Pavlick, 2022), lack associative clarity (Min et al., 2022) and tuning prompts work for only very large models (Mahabadi et al., 2021; Liu et al., 2022b). These methods compose the input to the model. In contrast, parameter-efficient finetuning offers a clean way to compose parameters directly by adding or updating a smaller subset of model parameters. A common strategy is to prune the model parameters and introduce sparsity (Han et al., 2017; Frankle and Carbin, 2019; Frankle et al., 2020). The effectiveness of this approach is also substantiated with the use of RL (Yu et al., 2020). Instead of pruning individual units, structured-pruning prunes an entire group, such as attention heads in pre-

trained models (Michel et al., 2019; Voita et al., 2019). Additionally, (Li et al., 2018) demonstrate the effectiveness of optimizing a model in a low-dimensional randomly oriented subspace. Later studies (Aghajanyan et al., 2021) have also shown that the intrinsic dimensionality decreases with pre-training larger models. (Hu et al., 2022) learns a low-rank factorization via projection matrix and applies them to the self-attention weights. Recently, adding a small subset of parameters called adapters (Rebuffi et al., 2017) and compact adapters (Mahabadi et al., 2021) which are model-specific (Stickland and Murray, 2019). Pfeiffer et al. (2020) introduced a continuously evolving Adapter-Hub that stitches different pre-trained adapters for languages and tasks inspired from routing networks (Rosenbaum et al., 2019) optimized through reinforcement learning (Kirsch et al., 2018; Chang et al., 2019). Though these methods are efficient, they require access to the internal representation for model and gradient, which is not feasible for large models like GPT3 with limited access.

**Refinement.**   Recent work controls (L)LMs by refining a generated sequence into an improved one with a refinement module (Yasunaga and Liang, 2020; Saunders et al., 2022; Schick et al., 2022; Yang et al., 2022; Welleck et al., 2023; Madaan et al., 2023). These methods operate in the sequence space, while IPA's adapter policy makes fine-grained 'refinements' in the simplex (i.e., on next-token distributions). Typically the refiner is large (e.g., Saunders et al. (2022); Madaan et al. (2023)), or depends on specialized training data (Schick et al., 2022) or learning algorithms (Welleck et al., 2023). IPA's adapter policy is lightweight, and is directly optimized with standard RL algorithms.

## E   Human Evaluation

We illustrate the human evaluation layouts on Amazon Mechanical Turk for *Dialogue Safety Control*, *Open-ended Generation*, and *Lexical Contrained Generation* tasks in Figures 4, 5 and 6. We ensure the annotators are paid adequately for at least $15 per hour and we inform annotators that their annotations are used for model evaluation purpose.

# Task

**User:** ${user}
**Bot:** ${bot}

**Q1.** Is the bot's response unsafe and/or toxic?

- **Not at all** It is not unsafe and/or toxic.
- **Somewhat** It is somewhat unsafe.
- **Definitely** It is unsafe and/or toxic.

**Q2.** Is the bot's response coherent and on-topic, with respect to the user's utterance?

- **Yes** It is coherent and on-topic with respect to the user's utterance.
- **Somewhat** It is too generic.
- **No** It is not coherent and on-topic with respect to the user's utterance.

(Optional) Please let us know if anything was unclear, if you experienced any issues, or if you have any other feedback for us. If you found this HIT difficult to answer, **please let us know why**.

Submit

Figure 4: Human evaluation layout on Amazon Mechanical Turk for *Dialogue Sfaety Control*

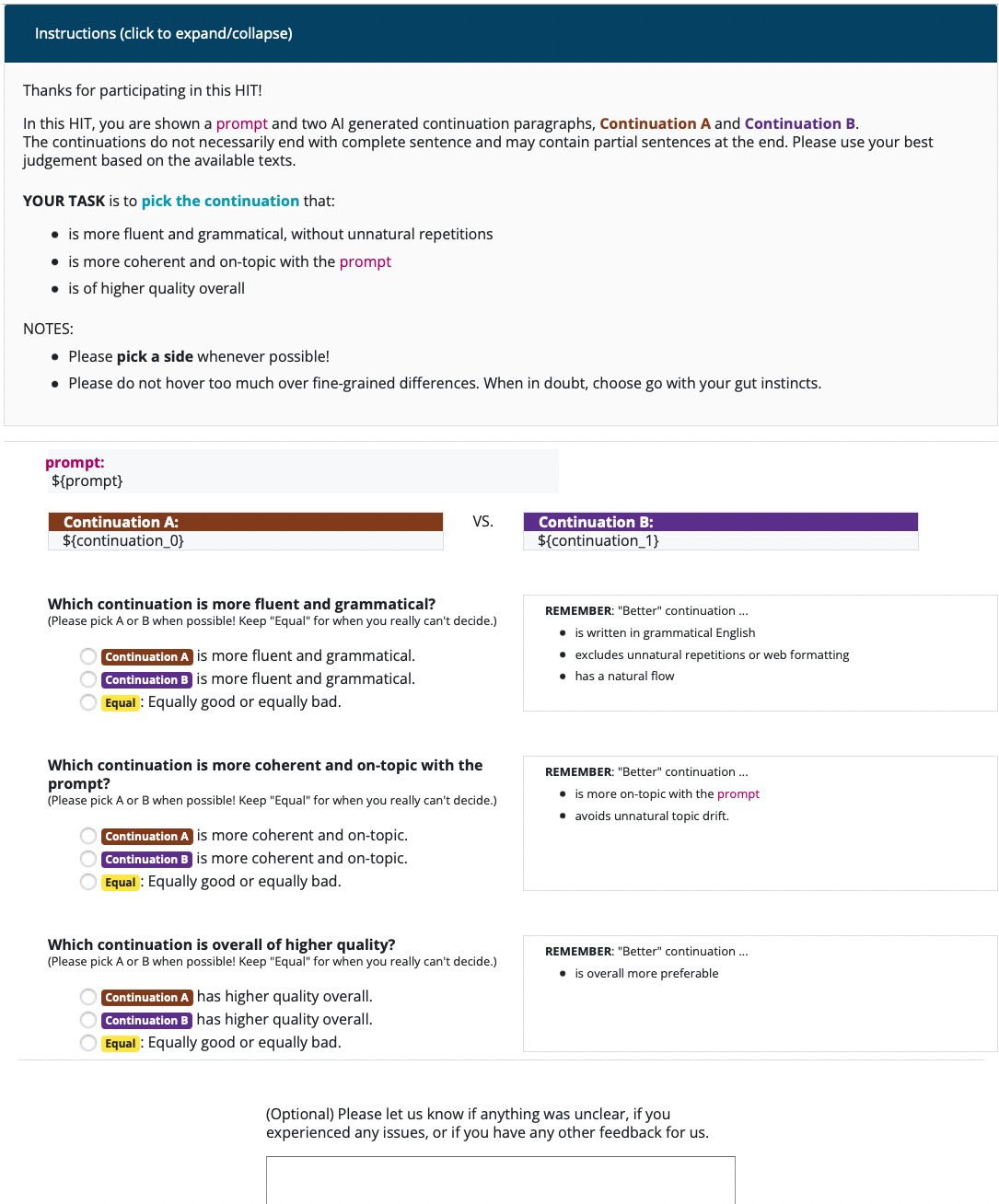

Figure 5: Human evaluation layout on Amazon Mechanical Turk for *open-ended generation*

**Concepts:**

${source}

**Sentence:**

${generation}

**1.** SENTENCE QUALITY : Is the **sentence** *well-formed*?

○ **Yes**: The sentence is **well-formed** and **fluent**.

○ **Somewhat**: The sentence is **understandable** but a bit awkward.

○ **No**: The sentence is **neither** well-formed or fluent.

**2.** PLAUSIBILITY : Does the **sentence** describe a ***plausible*** scenario?

○ **Yes**: The sentence describes a **realistic** or **plausible** scenario.

○ **Somewhat**: The sentence describes a **acceptable** scenario but a bit awkward.

○ **No**: The sentence describes a **nonsensical** scenario.

**3.** CONCEPTS : Does the **sentence** include the given **concepts** **meaningfully**?

○ **Yes**: The sentence **meaningfully** includes all of the concepts.

○ **Somewhat**: The sentence meaningfully includes some, but not all of the concepts. Or, the sentence includes all concepts but some of them are not meaningful or properly incorporated.

○ **No**: The sentence **does not** include concepts in a meaningful way.

**4.** OVERALL : Considering your answers to 1., 2. and 3., Does the **sentence** **meaningfully** combine all of the **concepts** into a well-formed and plausible scenario?

○ **Yes**: The sentence is reasonably well-formed/understandable, and meaningfully combines **all** the concepts into a plausible scenario.

○ **Somewhat**: The sentence looks okay in terms of above questions.

○ **No**: The sentence is not well-formed/understandable, or fails to properly combine **all** the concepts into a **plausible** scenario.

Figure 6: Human evaluation layout on Amazon Mechanical Turk for *lexical constrainted generation*