# OpenReview forum: "Inference-Time Policy Adapters (IPA): Tailoring Extreme-Scale LMs without Fine-tuning"
_EMNLP/2023/Conference — EMNLP 2023 Main_

### Official Review · Reviewer_pKXZ · 2023-07-25

**Typos Grammar Style And Presentation Improvements:** Very good writing
**Soundness:** 3

**Excitement:**

4: Strong: This paper deepens the understanding of some phenomenon or lowers the barriers to an existing research direction.

**Missing References:**

N/A

**Paper Topic And Main Contributions:**

The paper presents Inference-time Policy Adapters (IPA), an novel method to efficiently customize large language models like GPT-3 without expensive fine-tuning. Employing a lightweight policy adapter, IPA optimizes user objectives with reinforcement learning, enhancing control over LMs during decoding time. The paper demonstrates IPA's superiority over baseline methods and its proficiency in various text generation tasks, like toxicity reduction and lexically constrained generation.

**Questions For The Authors:**

1. How is the reward function for reinforcement learning chosen?

2. Is the same dataset split was used for both reinforcement learning and evaluation?

3. Proposed new experiment: use RL to optimize for one metric, and see if it hurts the performance with another metric in the same task. This experiment would help address concerns on whether such reward-oriented RL approach will hurt the generation quality not captured by existing metrics.

**Reasons To Accept:**

1. The paper addresses a crucial issue in generative large language models: the extensive computational resources required for fine-tuning. By incorporating a lightweight adapter during inference, the proposed method obviates the need to alter existing LLM parameters, thereby offering an efficient solution.

2. The design of the adapter and the reinforcement learning algorithm is very straightforward, making the proposed method simple yet effective

3. The method has been comprehensively tested on a variety of text generation tasks, where it consistently exhibits performance enhancements over existing baseline models.

**Reasons To Reject:**

1. The benchmarks and metrics used for evaluation appear to be subjective and potentially arbitrary. I understand that generative tasks lacks of well-known, gold-standard metrics such as BLEU, ROUGE, or F1/Accuracy. But I still find it challenging to compare against off-the-shelf baselines, diminishing confidence in the efficacy of the proposed method.

2. The algorithm requires manual configuration of the reward function for each task, e.g., calling Perspective APIs for toxicity reduction, or setting the *product* of coverage and fluency scores for lexically constrained generation. This approach seems arbitrary, with no principled methodology provided.

3. Given that the same metric is used for model evaluation, there's concern that the model might be overfitting to these metrics, rather than genuinely improving generation quality. Again, the metrics for these tasks does not include all aspects, such as LLM's "emergent abilities". It would be advisable to study whether using IPA aiming to improve some metrics will hurt the performance evaluated by a different set of metrics under the same task. Also, it is unclear whether the same dataset split was used for both reinforcement learning and evaluation. If so, serious concerns of overfitting will arise, and the method should be directly compared with supervised finetuning methods instead.

**Reproducibility:**

3: Could reproduce the results with some difficulty. The settings of parameters are underspecified or subjectively determined; the training/evaluation data are not widely available.

**Reviewer Confidence:**

4: Quite sure. I tried to check the important points carefully. It's unlikely, though conceivable, that I missed something that should affect my ratings.

---

> ### Author Rebuttal · Authors · 2023-08-29
>
> We thank reviewer pKXZ for their feedback! We’re thrilled to hear that they found our paper “addresses a crucial issue” and“ offering an efficient solution”. We’re also excited to learn they consider our proposed method as “simple yet effective”, “comprehensively tested on a variety of text generation tasks” and “consistently exhibits performance enhancements”.
>
> ### Re: the benchmarks and metrics used for evaluation appear to be arbitrary, challenging to compare against off-the-shelf baselines
> We respectfully disagree based on the previous literature.
>
> **The benchmarks we selected are widely used in previous literature** for controllable generations, and **represent diverse tasks across different domains**. RealToxicityPrompts is widely employed to evaluate toxicity reduction [1, 2, 4, 5, 9], CommonGen is extensively adopted to access hard lexical constraint satisfaction [5, 6, 7, 8, 9], WoW is widely used to gauge knowledge-grounded dialogue generation [12, 13, 14], DiaSafety is frequently adopted to test dialogue safety control [15, 16], and New articles are broadly employed to evaluate open-ended generation [3, 10, 11].
>
> Similarly, we closely followed previous works and **used metrics that are standard and widely used for the chosen benchmarks**. We also specified the previous work we followed for the experimental setups (see line 291, line 387, line 331, line 529, line 457). Importantly, **we reported multiple metrics for each task**, to measure task performance as well as language quality such as fluency, coherence, diversity (see Table 1, Table 3, Table 4, Table 5, Table 6). We did this to ensure that IPA genuinely improves generation quality rather than overfitting to a specific metric. Additionally, **we conducted human evaluation** to further assure generation quality (see Table 3, Figure 3, Table 5, Table 7). We observe IPA constantly outperforms all baselines across various automatic and manual metrics for all the tasks.
>
> As for baselines, we not only compare with the off-the-shelf models **but also existing SOTA methods for all the tasks**, to ensure the efficacy of our proposed method.
>
> ### Re: the algorithm requires manual configuration of the reward function for each task
> Yes and No.
>
> Yes in the sense that defining a reward function for a specific task is **the standard practice in the Reinforcement Learning (RL) field**, and a common procedure adopted across all RL literature.
>
> No in the sense that **we used the most commonly adopted reward function in previous literature** for each task. For example, PerspectiveAPI score has been widely used in numerous prior works [1, 2, 4, 5, 9] as a reward function for detoxification. Similarly, the product of coverage and fluency scores has been employed as a reward function for lexically constrained generation in various previous papers [5, 7, 8]. In fact, there is a line of work in the RL field studying the selection and improvement of reward function in various settings, such as Ramamurthy et al, 2022 [17], but that’s out of scope for our current work.
>
> Additionally, we conducted ablations with different reward functions for the same task in the appendix A.3. And we found IPA is highly effective with various reward functions.
>
> ### Re: the same metric is used for model evaluation
> We acknowledge that reward hacking is an important problem. To address this, **we've reported multiple metrics for the same task** in our original draft, **not only the one used as the reward function**. We measured task performance as well as language quality such as fluency, coherence, diversity (see Table 1, Table 3, Table 4, Table 5, Table 6). We did this to ensure that IPA genuinely improves generation quality rather than overfitting to a specific metric.
>
> Additionally, **we conducted human evaluation** to further assure generation quality (see Table 3, Figure 3, Table 5, Table 7). We observe IPA constantly outperforms all baselines across various automatic and manual metrics for all the tasks.
>
> As for exploring LLMs "emergent abilities", that does not align with the objective of the current paper and might lead us away from the core focus of our paper which is tailoring a large language model at decoding-time toward desired objectives without the need of fine-tuning.
>
> ### Re: selection of the reward function
> **We use the most commonly adopted reward function in previous literature** for each task. For example, PerspectiveAPI score has been widely used in numerous prior works [1, 2, 4, 5, 9] as a reward function for detoxification. Similarly, the product of coverage and fluency scores has been employed as a reward function for lexically constrained generation in various previous papers [5, 7, 8].
>
> ### Re: dataset split
> We only use the train and validation set in RL training. We confirm that the **test set is untouched and reserved for testing only**, and that all reported numbers are on the test set. We followed the standard train/val/test splits for all the tasks.
>
> ### Re: proposed new experiment
> **We’ve already included the experiment suggested by the reviewer in our original draft**. For each task we report multiple metrics, including those that are not part of the reward function. More specifically, we measure task performance as well as language quality such as fluency, coherence, diversity  (see Table 1, Table 3, Table 4, Table 5, Table 6) and found IPA constantly improves task performance while maintaining generation quality.
>
>
>
> #### **Bibliography**
> [1] Liu, Alisa et al. “DExperts: Decoding-Time Controlled Text Generation with Experts and Anti-Experts.” Annual Meeting of the Association for Computational Linguistics (2021).
>
> [2] Lu, Ximing et al. “Quark: Controllable Text Generation with Reinforced Unlearning.” ArXiv abs/2205.13636 (2022): n. Pag.
>
> [3] Li, Xiang Lisa et al. “Contrastive Decoding: Open-ended Text Generation as Optimization.” Annual Meeting of the Association for Computational Linguistics (2022).
>
> [4] Krause, Ben et al. “GeDi: Generative Discriminator Guided Sequence Generation.” Conference on Empirical Methods in Natural Language Processing (2020).
>
> [5] Welleck, Sean et al. “Generating Sequences by Learning to Self-Correct.” ArXiv abs/2211.00053 (2022): n. pag.
>
> [6] Qin, Lianhui et al. “COLD Decoding: Energy-based Constrained Text Generation with Langevin Dynamics.” ArXiv abs/2202.11705 (2022): n. Pag.
>
> [7] Lu, Ximing et al. “NeuroLogic Decoding: (Un)supervised Neural Text Generation with Predicate Logic Constraints.” ArXiv abs/2010.12884 (2020): n. Pag.
>
> [8] Lu, Ximing et al. “NeuroLogic A*esque Decoding: Constrained Text Generation with Lookahead Heuristics.” North American Chapter of the Association for Computational Linguistics (2021).
>
> [9] Kumar, Sachin et al. “Gradient-based Constrained Sampling from Language Models.” Conference on Empirical Methods in Natural Language Processing (2022).
>
> [10] Su, Yixuan et al. “A Contrastive Framework for Neural Text Generation.” ArXiv abs/2202.06417 (2022): n. Pag.
>
> [11] Meister, Clara et al. “Typical Decoding for Natural Language Generation.” ArXiv abs/2202.00666 (2022): n. Pag.
>
> [12] Lewis, Patrick et al. “Retrieval-Augmented Generation for Knowledge-Intensive NLP Tasks.” ArXiv abs/2005.11401 (2020): n. Pag.
>
> [13] Shuster, Kurt et al. “Retrieval Augmentation Reduces Hallucination in Conversation.” Conference on Empirical Methods in Natural Language Processing (2021).
>
> [14] Dziri, Nouha et al. “FaithDial: A Faithful Benchmark for Information-Seeking Dialogue.” Transactions of the Association for Computational Linguistics 10 (2022): 1473-1490.
>
> [15] Sun, Hao et al. “On the Safety of Conversational Models: Taxonomy, Dataset, and Benchmark.” ArXiv abs/2110.08466 (2021): n. Pag.
>
> [16] Meade, Nicholas et al. “Using In-Context Learning to Improve Dialogue Safety.” ArXiv abs/2302.00871 (2023): n. pag.
>
> [17] Ramamurthy, Rajkumar et al. “Is Reinforcement Learning (Not) for Natural Language Processing?: Benchmarks, Baselines, and Building Blocks for Natural Language Policy Optimization.” ArXiv abs/2210.01241 (2022): n. pag.

---

### Official Review · Reviewer_44dF · 2023-08-03

**Soundness:** 3

**Excitement:**

3: Ambivalent: It has merits (e.g., it reports state-of-the-art results, the idea is nice), but there are key weaknesses (e.g., it describes incremental work), and it can significantly benefit from another round of revision. However, I won't object to accepting it if my co-reviewers champion it.

**Paper Topic And Main Contributions:**

This paper proposes Inference-time Policy Adapters (IPA), which efficiently tailors a language model such as GPT-3 without fine-tuning it. IPA guides a large base model during decoding time through a lightweight “policy adapter” trained to optimize an arbitrary user objective with reinforcement learning.

Good results were reported when tailoring a list of strong baseline models, such as GPT-2.


**Questions For The Authors:**

Questions:
1.	What motivated the usage of GPT-2 for a list of experiments? It is relatively small and just prefer to learn more sota models (e.g., LLaMA and related LLMs) and the related results.


**Reasons To Accept:**

Strong:
1.	The whole IPA idea is interesting and effective.
2.	Only inferencing time adapter will save a lot of time for model alignment during fine-tuning or peft.
3.	Impressive results were reported for tailoring gpt-2 which fit gpt-3’s results.


**Reasons To Reject:**

Weak:
1.	Model aligning will still be an important direction for LLM adaptation and a combined usage and related investigation will be required.


**Reproducibility:**

3: Could reproduce the results with some difficulty. The settings of parameters are underspecified or subjectively determined; the training/evaluation data are not widely available.

**Reviewer Confidence:**

4: Quite sure. I tried to check the important points carefully. It's unlikely, though conceivable, that I missed something that should affect my ratings.

---

> ### Author Rebuttal · Authors · 2023-08-29
>
> We thank reviewer 44dF for their feedback! We’re excited that they found our IPA is “an interesting and effective idea", with “impressive results” and “will save a lot of time for model alignment”.
>
> ### Re: Reason to reject
> We would appreciate it if reviewer 44dF could clarify their reason to reject. We’re **puzzled** by the reason to reject as it appears to be a **positive statement** rather than a valid justification for rejection. We do agree that “model aligning will still be an important direction” and that “related investigation will be required”. As a response, our paper investigates efficient strategies for aligning large-scale language models through the use of an inference-time policy adaptor.
>
> ### Re: motivation for the usage of GPT-2
> First, we’d like to remind the reviewer that **we’ve also included the most powerful large-scale models**, such as **GPT3**, **Blenderbot**, both as a model to tailor and as a baseline in the experiments for *all the tasks*. (see Table 1, Table 3, Table 4, Table 5, Table 6). As we highlighted in the paper (see line 309, line 368, line 439, line 485, line 569), we observed a significant performance gain on top of these powerful models brought by IPA, which surpassed the existing SOTA baselines across all the tasks.
>
> Second, we choose GPT-2 in our experiments to **showcase the effectiveness and efficiency of IPA**. We demonstrate that **a small language model, i.e GPT2-large, as the policy adaptor can successfully tailor a x200 larger base language model GPT3** across various tasks (see line 309, line 368, line 439). We also show that **directly applying the policy adapter optimized for GPT-2 on top of GPT-3 (IPA-) is highly effective**, showcasing the adaptability and reusability of IPA (see line 308, line 438). Both aspects highlight the lightweight and easily accessible nature of our approach, making it suitable for the wider NLP community, particularly in limited resource settings.
>
> Finally, **our method is orthogonal to the models used**. As mentioned earlier, we’ve illustrated the effectiveness of IPA with multiple most powerful large-scale language models in the original draft. Nonetheless, we **added additional experiments with LLaMA models** for constrained generation tasks as suggested. We apply IPA to tailor an off-the-shelf LLaMA-13B model and initialize the policy adapter with a LLaMA-7B model. As shown in table below, IPA leads to remarkable improvement on top of LLaMA-13B in terms of constraint coverage while maintaining language quality. We’ll add LLaMA experiments for other tasks as well in the final draft.
>
> | Models   |      Coverage      |  Fluency |
> |----------|:-------------:|:------:|
> | LLaMA-13B |  28.73 | 89.64 |
> | IPA (LLaMA-13B) |   81.49   |   89.71 |

---

### Official Review · Reviewer_Ge8N · 2023-08-04

**Soundness:** 4

**Excitement:**

4: Strong: This paper deepens the understanding of some phenomenon or lowers the barriers to an existing research direction.

**Paper Topic And Main Contributions:**

This paper proposes Inference-time policy adapters, a new method to efficiently adapt large language models to user specified objective via reinforcement learning. More specifically, a small policy adaptor is trained to adjust the (frozen) base LM output distribution while optimizing a pre-defined reward signal for a specific task. Experimental results on five text generation tasks confirm the efficacy of the proposed method.

**Reasons To Accept:**

1. This paper is well-written.
2. The research problem is well-motivated and relevant to many researchers studying large language models.
3. The experiments and evaluations are thorough and promising.

**Reasons To Reject:**

When the extreme-scale policy model is not feasible to use and when a smaller model from the same model family is not present, extra computation is needed to derive the approximate policy.

**Reproducibility:**

3: Could reproduce the results with some difficulty. The settings of parameters are underspecified or subjectively determined; the training/evaluation data are not widely available.

**Reviewer Confidence:**

3: Pretty sure, but there's a chance I missed something. Although I have a good feel for this area in general, I did not carefully check the paper's details, e.g., the math, experimental design, or novelty.

---

> ### Author Rebuttal · Authors · 2023-08-29
>
> We thank reviewer Ge8N for their encouraging feedback! We’re thrilled to hear that they found our paper is “well-motivated”, “well-written”, “relevant to many researchers studying large language models” and our “experiments and evaluations are thorough and promising”.
>
> ### Re: “extra computation is needed to derive the approximate policy”
> First, we’d like to clarify that **the computation of distilling an approximate policy is in fact very efficient**. As we highlighted in the original draft (**line 374-376**): “while fine-tuning GPT-3 costs \\$156.82, training a distilled GPT-3 as the approximate policy requires only \\$28.59 for generating outputs from GPT-3”. In terms of time and compute cost, it takes around 3 GPU hours to train a distilled GPT-3 on 8 NVIDIA Quadro 1394 RTX 8000 GPUs. We’ll include this additional detail in our revision.
>
> Second, in the case of recently released *open source* large language models, **there is always a smaller model within the same model family** that can be directly used as an approximate policy. We surveyed over multiple recent model families, including Flan-T5, PaLM2, PaLM, Falcon, LLaMA, LLaMA2, Vicuna, MPT, Lazarus, WizardLM, GPT4All, and found all of them release a series of models of various sizes.

---

### Meta-Review · Area_Chair_3ikC · 2023-09-07

**Recommendation:** 5

**Metareview:**

This work presents IPA (Inference-Time Policy Adapters), a novel way of adapting LLMs to various generation tasks by "tailoring" their output distribution through task-specific training of other (smaller) LMs via reinforcement learning (RL).

The output probability for the next token prediction is modelled as the normalized product of the original and adapter LM probabilities. Crucially, during training, the LLM is frozen, and only the adapter is updated with the RL process.

The efficacy of IPA is demonstrated on 5 generation tasks using various SOTA LLMs as base models, and smaller variants or distilled LMs as adapters. For each task relevant RL rewards are devised, IPA is trained, and automatic and human evaluation is carried out, against a variety of strong baselines.\
IPA proves to be effective on all tested tasks, notably sometimes even when a nominally less capable model (GPT2) is used to tailor a strong LLM (GPT3).

Some ablations wrt. reward modelling are presented (Appendix), showing that IPA is somewhat robust to the precise reward, yet performs best when the full proposed rewards are used.

## Strengths
Reviewers agree the paper is sound (4, 3, 3) and exciting (4, 4, 3), and I agree with the positive aspects mentioned by reviewers.
- The paper is very well written and easy to follow; it is clearly structured, and presents the proposed novelties in a concise and understandable manner.
- The topic of adapting (extremely) large LLMs is very current, and the authors propose a novel method that seems both easy to implement and train, and effective on a variety of tasks.
- The conducted experiments are extensive and seem to support the author's claims; the use of human evaluation for the presented generation tasks is commendable and largely supports the results of automatic metrics.

## Weaknesses
Concerns were raised wrt. to various aspects of the proposed method. The authors provided extensive rebuttals and addressed, in my opinion, all relevant criticism in a satisfactory manner.
- Availability of smaller versions of base LLMs might be an issue; however, it is demonstrated that (i) often smaller versions are indeed available; (ii) distilled versions can be derived relatively cost-efficient; (iii) from the paper's results, even not directly related models might be viable candidates for adapter policies.
- Some of the chosen benchmarks/metrics were questioned; I find myself agreeing with the authors that the these are standard and relevant.
- The need for task-specific rewards was criticised. I again have to agree with the authors that such task-specific rewards are inherent to the RL process. While the choice of specific rewards in the paper could have been better motivated in the description of each task to make the it more understandable, the individual rewards are (in my opinion) sensible and justified by the tasks and relevant related work; clarification is an easy one-line fix per task for a camera ready version.
- Related, it was pointed out that directly optimising a reward that is also the test evaluation metric is potentially problematic; however, experiments are extensive enough to show improvements persist for metrics that are not used for reward modelling, as well as for human evaluation.

## Conclusion
The authors present a strong piece of work that is well-motivated, relevant, and extensively evaluated. In my opinion, IPA constitutes a valuable addition to the growing toolkit of "parameter-efficient" methods for LLM adaptation. In particular, IPA is applicable in settings where the underlying LM's parameters are not accessible, and only its output distribution is observable.

The efficacy of the proposed approach is demonstrated clearly on 5 different generative tasks, through automatic and human evaluation.\
I only have three small points of criticism, mostly wrt. evaluation:
- The human evaluation results of 4.2 (Lexically Constrained Generation) seem surprisingly weak compared to the clear improvements on automatic metrics. Looking at the Mechanical Turk question template I wonder if a 3-point Likert scale and the measured aspects are the right tool here, as even the base GPT-3 achieves 2.60 on the "Overall" question, and even higher on "Quality" and "Plausibility". Still, IPA improves over these base model scores.
- Task 4.5 (Knowledge-grounded Dialog) is the only one that was not put to a human evaluation; there might be reasons, which could briefly be pointed out.
- The title "[...] without Fine-tuning" might be construed to imply that no training is happening at all (e.g., that IPA is purely inference-based). This is obviously not the case, as the adapter policy is indeed trained (one could say with task-specific fine-tuning) via reinforcement learning. However, the abstract clarifies the true nature of the approach.

None of these points seem severe enough to not merit acceptance, and the work in itself should make a valuable contribution to the main conference.

---

### Decision · Program_Chairs · 2023-10-07

**Decision:**

Accept-Main

**Comment:**

This work presents IPA (Inference-Time Policy Adapters), a novel way of adapting LLMs to various generation tasks by "tailoring" their output distribution through task-specific training of other (smaller) LMs via reinforcement learning (RL).

The output probability for the next token prediction is modelled as the normalized product of the original and adapter LM probabilities. Crucially, during training, the LLM is frozen, and only the adapter is updated with the RL process.

The efficacy of IPA is demonstrated on 5 generation tasks using various SOTA LLMs as base models, and smaller variants or distilled LMs as adapters. For each task relevant RL rewards are devised, IPA is trained, and automatic and human evaluation is carried out, against a variety of strong baselines.\
IPA proves to be effective on all tested tasks, notably sometimes even when a nominally less capable model (GPT2) is used to tailor a strong LLM (GPT3).

Some ablations wrt. reward modelling are presented (Appendix), showing that IPA is somewhat robust to the precise reward, yet performs best when the full proposed rewards are used.

## Strengths
Reviewers agree the paper is sound (4, 3, 3) and exciting (4, 4, 3), and I agree with the positive aspects mentioned by reviewers.
- The paper is very well written and easy to follow; it is clearly structured, and presents the proposed novelties in a concise and understandable manner.
- The topic of adapting (extremely) large LLMs is very current, and the authors propose a novel method that seems both easy to implement and train, and effective on a variety of tasks.
- The conducted experiments are extensive and seem to support the author's claims; the use of human evaluation for the presented generation tasks is commendable and largely supports the results of automatic metrics.

## Weaknesses
Concerns were raised wrt. to various aspects of the proposed method. The authors provided extensive rebuttals and addressed, in my opinion, all relevant criticism in a satisfactory manner.
- Availability of smaller versions of base LLMs might be an issue; however, it is demonstrated that (i) often smaller versions are indeed available; (ii) distilled versions can be derived relatively cost-efficient; (iii) from the paper's results, even not directly related models might be viable candidates for adapter policies.
- Some of the chosen benchmarks/metrics were questioned; I find myself agreeing with the authors that the these are standard and relevant.
- The need for task-specific rewards was criticised. I again have to agree with the authors that such task-specific rewards are inherent to the RL process. While the choice of specific rewards in the paper could have been better motivated in the description of each task to make the it more understandable, the individual rewards are (in my opinion) sensible and justified by the tasks and relevant related work; clarification is an easy one-line fix per task for a camera ready version.
- Related, it was pointed out that directly optimising a reward that is also the test evaluation metric is potentially problematic; however, experiments are extensive enough to show improvements persist for metrics that are not used for reward modelling, as well as for human evaluation.

## Conclusion
The authors present a strong piece of work that is well-motivated, relevant, and extensively evaluated. In my opinion, IPA constitutes a valuable addition to the growing toolkit of "parameter-efficient" methods for LLM adaptation. In particular, IPA is applicable in settings where the underlying LM's parameters are not accessible, and only its output distribution is observable.

The efficacy of the proposed approach is demonstrated clearly on 5 different generative tasks, through automatic and human evaluation.\
I only have three small points of criticism, mostly wrt. evaluation:
- The human evaluation results of 4.2 (Lexically Constrained Generation) seem surprisingly weak compared to the clear improvements on automatic metrics. Looking at the Mechanical Turk question template I wonder if a 3-point Likert scale and the measured aspects are the right tool here, as even the base GPT-3 achieves 2.60 on the "Overall" question, and even higher on "Quality" and "Plausibility". Still, IPA improves over these base model scores.
- Task 4.5 (Knowledge-grounded Dialog) is the only one that was not put to a human evaluation; there might be reasons, which could briefly be pointed out.
- The title "[...] without Fine-tuning" might be construed to imply that no training is happening at all (e.g., that IPA is purely inference-based). This is obviously not the case, as the adapter policy is indeed trained (one could say with task-specific fine-tuning) via reinforcement learning. However, the abstract clarifies the true nature of the approach.

None of these points seem severe enough to not merit acceptance, and the work in itself should make a valuable contribution to the main conference.